# Sense of Place: Narrating Emotional Experiences of Malaysian Borneo through Western Travel Blogs

Siao Fui Wong [1,*] , Balvinder Kaur Kler [2,*] and Bamini KPD Balakrishnan [3,*]

1   MODUL School of Tourism and Hospitality Management Nanjing, Nanjing Tech University Pujiang Institute, Nanjing 211222, China
2   Borneo Tourism Research Centre (BTRC), Faculty of Business, Economics and Accountancy, Universiti Malaysia Sabah, Kota Kinabalu 88400, Malaysia
3   Faculty of Business, Economics and Accountancy, Universiti Malaysia Sabah, Kota Kinabalu 88400, Malaysia
*   Correspondence: wongsiaofui@njpji.cn (S.F.W.); balvinder@ums.edu.my (B.K.K.); bamini@ums.edu.my (B.K.B.); Tel.: +86-15951020771 (S.F.W.); +60-11-2602-5682 (B.K.K.); +60-11-5111-3234 (B.K.B.)

**Abstract:** Tourists' sense of place or destination attachment could play an important role in destination branding. Yet, sense of place literature focuses on residents as the concept originates from a long-term residence in one place. This study explores the role of destination attachment based on tourist experience for branding based on a case study in Malaysian Borneo. A qualitative content analysis using QCAmap of 34 blogs extracted 116 blogposts with narratives containing emotional sentiments from international tourists. Findings uncovered six important attributes: namely nature, adventure, environment, culture, conservation, and education associated with destination attachment. Findings show that these attributes are interconnected as the main attribute, nature, produces other attributes. A tourists' sense of place model for destination attachment is proposed to understand how tourists develop attachment to a place.

**Keywords:** sense of place; blog; Borneo; emotional experiences; QCAmap





## 1. Introduction

Destination branding is a marketing process that aims to enhance tourists' awareness of the destination and its brand [1]; this process should involve both residents and tourists. A resident-centred approach considers locals as the co-creators and co-producers of authentic destination brand identity [2–5] while a tourist-centred approach creates a positive destination brand image based on tourists' perceptions and experiences [6]. Brand identity is how the destination management organisations (DMOs) want the destination value to be perceived by tourists [7] and brand image is the actual perception of tourists towards a destination brand [8]. Regardless, identity and image are derived from an individual's interactions with a place. Scholars agree that resident and tourist experiences with a destination produce a sense of place (SoP), emotional bonds based on an individual's attachment to the destination [9]. Understanding and utilizing resident SoP creates a unique local identity ensuring a sustained destination brand [4]. Understanding tourist SoP can help DMOs elevate specific attributes tourists are emotionally attached to for branding destinations to specific market segments [10].

SoP is often understood as a strong place identity unique to the residents. Residents' SoP is co-created through social constructions and reproductions of everyday life influenced by cultural and historical links [2]. As residents mutually construct lived experiences "in" and "with" the place they live in [11], these produce place meanings which constitute a unique SoP [2]. According to Hay et al. [4], residents express place identity through place stories and articulate an authentic destination brand identity. Literature on resident's SoP often explores locals' perceptions of a destination brand and their destination ambassador

behaviour [4]. Although residents' SoP produces cultural and social significances [4], tourists' perceived image alone is sufficient to create brand awareness [12]. For this reason, Hay et al. [4] called for studies on how other stakeholder groups can challenge the role of residents in the destination branding process. At present, destination brand development models emphasise tourist-centred approaches, but it remains unclear which models could be considered effective in managing destination branding [13].

Tourist SoP is on the other end of the destination branding continuum. Tourists will reject a destination brand if it lacks local identity [14]. When a strong place identity is instilled in a destination brand, it triggers emotions and evokes strong attachments to a place [15]. Tourists are willing to participate in the destination branding process and are looking for greater involvement with the destination and the brand itself [13]. Whose SoP would better simulate an authentic destination image to other tourists if not the emotional experiences embedded in the perceived image of the tourists themselves? Emotional experience is a feeling shaped by individual's evaluation of places [16]. Tourists' emotions, connections, and affection toward a destination produces SoP and produces emotional attachment to a destination and its brand [17]. Like residents, tourist SoP contains emotional experiences that evoke other tourist's emotional engagement in blogs [18]. Emotional engagement refers to the forming of an emotional relationship between tourists and a place [18]. Tourists may unintentionally produce SoP when sharing their stories. Tourists make sense of places through recollection and emotional experiences in their narratives [19] and project their SoP in the form of perceived images. Therefore, it is proposed that tourists' SoP is rooted in narratives [20].

In the era where user-generated content (UGC) influence is strong, tourists' SoP narrated through destination images shared on social media sites not only enhances destination brand awareness [21], but it also evokes emotional engagement since it allows potential tourists to experience stories from first-person perspectives [18]. Social media provides meaningful data [22] and allows researchers to enhance place-based management and branding [10] (p. 84). However, research that explores the emotional bonds between social media users and tourist destinations is limited [23]. Particularly, travel blogs that contain a wide range of emotions [24] that are accessible to potential tourists [25] receive less attention in academia compared to other types of social media such as Facebook, Twitter, YouTube, and TripAdvisor [26]. Blogs contain subjective interpretations of tourists' own experiences [24] and raw emotions that could evoke emotional engagement among other tourists [18]. How these emotional experiences embedded through tourists' perceived images evoke emotional engagement among other tourists has not been studied, as the existing literature is dominated by quantitative research methods that limit the potential to understand tourists' thoughts and feelings [25]. Analysing blogs using a qualitative approach not only provides an understanding on what attracts tourists to a destination, but also shows how events are transformed into meaningful experiences [27] and produce emotional engagement for potential tourists before visitation. Therefore, this study explores tourist perceived images through a qualitative content analysis (QCA) on blogposts to better understand tourist SoP.

Before presenting the rationale for this study, the paper presents a brief background of the case study site in Malaysian Borneo. The site is chosen because it is a destination with a strong historical presence in European literature. The influence of early literature on Western tourists is strong as these tourists may develop an emotional experience with the site before travel. In this case, tourists perceive Sabah, Malaysian Borneo as "Wild Borneo" and expect to see what is told upon arrival [28]. Next, the literature review focuses on SoP for branding, uncovering SoP through blogs. Thirdly, the methods section clarifies the use of a qualitative content analysis map (QCAmap) to explore the narratives written by bloggers for a case study site to reveal tourists' SoP. Data collection and data analysis are delineated followed by findings, discussion, conclusions, and potential future work. This study contributes a conceptual model for destination branding using tourist SoP.

*1.1. The Case—Sabah, Malaysian Borneo*

Sabah in Malaysian Borneo is situated by the South China Sea. Together with the Malaysian Borneo region of Sarawak, the Kingdom of Brunei, and Kalimantan, Indonesia, Nusantara in East Kalimantan is set to be the site of the new capital of Indonesia by 2024. Sabah is one of the four regions on the Borneo Island (see Figure 1) renowned for its image as a nature–adventure destination.

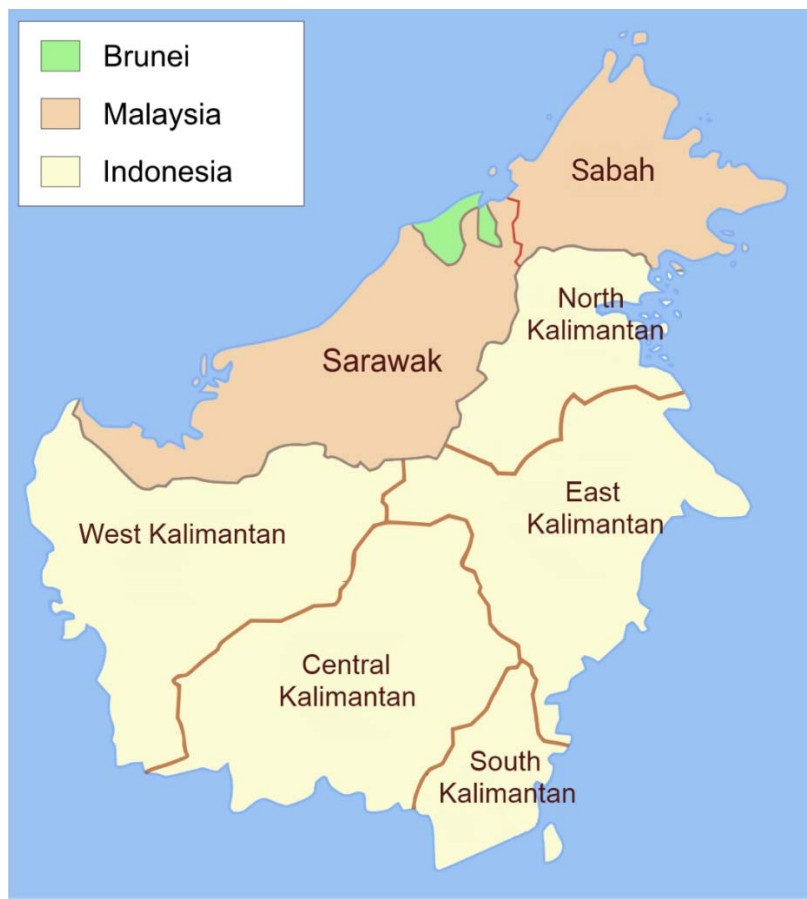

**Figure 1.** The island of Borneo. Source: Ranking Update, CC BY-SA 3.0.

Wild, untamed, unexplored, dangerous, mysterious, and exotic are the projected images binding the four political territories [29], and the famous orangutan is the 'embodiment of pristine nature' [30] (p. 72). From a Western perspective, these images are well-narrated in literature, books, documentaries, tourism promotional materials, and other printed and non-printed materials in three themes: nature, adventure, and culture [28,29,31–34]. "Nature" refers to the inherent features of Borneo such as wildlife, biodiversity, and unique landscapes; "adventure" is human-environment interactions denoting human-made activities, exploration, and experiences that include hard adventure, soft adventure, and exploration of culture, local way of life, and natural environment; and "culture" denotes to the manifestations of a collective group of people [32]. In particular, "nature" offers the opportunity to trek and explore the wilderness, which creates the theme of "adventure" and indigenous groups that leads to the theme of "culture" [35]. Figure 2 illustrates the link between the themes.

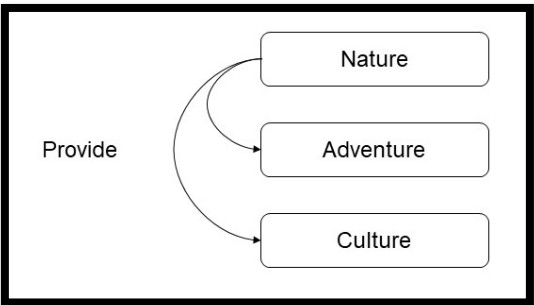

**Figure 2.** Images of Sabah, Malaysian Borneo projected in literature and tourist brochures.

### 1.2. Rationale of Study

Tourists expect to see images described in literature and tourist brochures [27]. Their expectations form meanings about a place before visitation [36]. In the pre-travel stage, tourists develop an induced image of the destination, which is formed in the second stage of the image-formation process [37]. The three-stage image-formation process suggests that perceived image is developed from unintended exposure to non-tourism information (organic image), to purposeful search of travel information (induced image), and to experience-based perceptions (modified-induced/complex image) [38]. Literature and tourist brochures produce an induced image that strengthens the image held in potential tourists' minds. These images, which are formed through stories of others [39], will allow potential tourists to develop emotional engagement even before visitation [36]. In the case of Sabah, Malaysian Borneo, tourists expect to see nature and explore adventure and culture, which are the induced SoP developed before visitation. After three decades, the images of nature, adventure, and culture remain unchanged in tourist brochures. However, different images of Sabah and Malaysian Borneo are projected in recent literature. These include the availability of good infrastructure, modern cities, and English-speaking locals [34], but also the negative environmental impacts that threaten the natural environment including activities that affect the wildlife, species extinction, oil palm plantations, and pollution [40]. In addition, consulting blogs before visitation is a common practice [41]. If these new images alter the existing induced image, how is tourist SoP for Sabah, Malaysian Borneo altered after visitation? As the combination of existing meaning with actual interactions further develops tourist understanding [39], how are tourists developing emotional attachments to a destination [42]? The next section reviews literature pertinent to SoP and discusses how SoP can be formed and shared through blogs.

## 2. Literature Review

### 2.1. Sense of Place

SoP is constructed from subjective and emotional people–place interactions [2]. It consists of three elements: namely place attachment, place identity, and place dependence [43]. Some scholars argue that place identity and place dependence are two elements of place attachment [44]. Others describe SoP as the interaction between place meaning and place attachment [45]. Regardless, SoP is how people assigned symbolic meanings (place meaning), emotional attachment (place identity), and functional attachment (place dependence) to a place based on their subjective and emotional interactions [46]. The concept of SoP, which is linked to conceptions of place attachment, topophilia, insidedness, and community sentiment, is the foundational concept that describes human–place interaction [47]. Therefore, SoP refers to bonds between people and places [48]. From a social psychology perspective, this bond differs across physical, psychological, emotional, and experiential elements and should be studied separately [49]. For example, the study of place attachment (the emotional element of SoP) would contribute to a deeper understanding of individual's emotional attachment to place [42].

SoP originates from long-term residence. Place attachment theory suggests that resident's SoP is a success factor for destination branding and marketing [50]. Target audiences will reject a destination brand that is not created based on the resident's SoP [14] because it will lack an authentic identity [51]. As residents co-create stories according to their rooted appreciation of the destination-based values and meanings [3], a destination brand that reflects changes in place meanings through resident's stories will communicate a more compatible local image [4]. Therefore, SoP is understood as a strong place identity uniquely produced by the residents in the destination branding literature [3–5]. However, tourists also produce SoP. Yuksel, Yuksel, and Bilim [52] emphasised tourist SoP as destination attachment with indications that the emotional meanings tourists attach to the places they visit influence satisfaction and loyalty. Qu, Dong, and Xiang [53] concur and found destination attachment does initiate and perpetuate revisits by loyal tourists by considering SoP through destination attachment as a non-habitual place for tourist. Their study demonstrated the attachment-triggering effects of critical attributes that affected tourists' revisit decision.

Selwyn [54] asserts that tourist's imagination constructs an image of a place. This image allows tourists to develop a SoP—meanings about a place even pre-travel [36]. For example, the image of Borneo conveyed in early literature and tourist brochures has shaped modern tourists' expectations about the island and its people [28]. As meanings are formed within tourists' knowledge, expectations, and imagination of a place, their SoP is developed depending on the level of exposure and interactions with the destination [39]. Tourists may develop different levels of emotional attachment in different stages of the interaction with destination. Regardless, tourists are emotionally attached to a destination and its brand as they develop emotions, connection, and affection towards the destination based on how they value a destination [17].

Two major dimensions used to describe a SoP are: (1) relationship to place, which refers to the various ways in which individuals are connected to places and the various sorts of bonds (genealogical/historical, narrative/cultural, economic, ideological, cosmological, and dynamic); and (2) the degree and forms of attachments to one location (place identification, place dependency, affective attachment, social bonding, place memory, and place expectation) are reflected in place attachment [55–57].

SoP is a multi-dimensional construct. Each dimension addresses different aspect of SoP. Studies explore the relationship between the constructs but the main concern among researchers is how individuals are emotionally attached to a place [42]. To fully operationalize SoP, Nelson et al. [42] suggest an exploration of the constructs through factors. That is, geography, resources, community, and environment that affect the people–place bond as a way to measure how individuals value a place, why they depend on the place and what is the meaning of place to them on a symbolic level (p. 256). At present, destination brand development models are emphasising tourist-centred approaches but remain unsure which model could be considered effective in managing destination branding [13]. Tourists' SoP allows potential tourists to imagine a destination from the first-person perspective. As such, a destination branding model based on tourist's SoP implies a strong destination image that triggers emotions and evokes emotional engagement between potential tourists and destinations [18]. Therefore, it is timely to explore how tourists' SoP contributes to the branding of destinations.

According to Jarratt, Phelan, Wain, and Dale [58], tourists' SoP features local distinctiveness. For example, Morecambe Bay of England is popular for its local history, natural environment, landscape, and experience such as beaches, wildlife or nature reserves. Whilst developing a SoP toolkit for Morecambe Bay tourists, Jarratt et al. [58] captured emotional sentiments according to these local features and revealed four tourists' SoP. These include wildlife and nature, landscape and views, culture and heritage, and food and drink. These aspects may not be applicable to tourists who visit other places. However, these unique buying points, if identified and communicated through emotional sentiments of the place, will become unique selling points which ultimately promote the

localness of a place. Tourists' SoP is uniquely formed within time and space. In the case of protected areas in Spain and Chile, tourists' SoP and meaning is understood based on their experiences in nature-based tourism destinations [59]. This SoP model may only be applicable for similar types of destinations.

Jarratt et al. [58] emphasised that the fluid nature of places causes the dynamism of SoP and place brand. Although a few SoP models are developed from the perspectives of residents [2–4], there is no fixed model that frames the components of tourists' SoP. Though it can be guided by place-related themes, place specific facts, research and recommendation on visitor perceptions, branding guidelines, and potential networking information [58], tourists' SoP is uniquely constructed based on their subjective and emotional interaction with each place. This is why a unified tourists' SoP model is not available in the literature. Tourists' SoP differs across experiences and changes as experience changes. One way to capture tourists' SoP is to reflect distinctive local features based on their emotional sentiments [58]. McCreary et al. [10] suggest the possibility to assess tourists' SoP through UGC, which enables a more tourist-centred destination image. How tourists' SoP are emotionally attached to a destination brand is of concern for scholars [42]. Recent growth of UGC predisposes blogs as purveyors of SoP, which is presented in the next section.

### 2.2. Understanding Tourists' Sense of Place through Blogs

Although a tourists' SoP model cannot be unified, SoP is rooted in narratives [20] and is deepened as tourists experience and recall the destinations through narratives [60]. Tourists' SoP can be explored through UGC projected in social media sites. McCreary et al. [10] suggest that the destination images generated by users contains tourists' SoP. For example, Feick and Robertson [61] discovered differences in local and global urban place descriptors through measuring photo tags in Flickr. Similarly, McCreary et al. [10] demonstrated how UGC provides meaningful social values useful for place-based management and branding in the North Shore of Minnesota. Social values are "collective attachment to place embodies meanings and values that are important to a community" [62] (p. 22). From this perspective, the collective attachment to a destination can be understood as a dynamic concept that involves various stakeholder perspectives and changes across time and space [63]. Meanings are co-constructed by all stakeholders including tourists. While DMOs lack control of how tourists construct place meanings, they capitalise on certain meanings and images perceived by tourists [64]. Moreover, tourist destination image evolves with time and space [65]. Constant evaluation of UGC not only refines destination marketing and promotion strategies, but it also improves relationships between DMOs and tourists [13]. More importantly, UGC displays tourists' subjective evaluation and emotional attachment to places [23]. Some images, although not usually reflected by the DMOs, are topics of interest of tourists [66] and the most authentic perceived by the others. Therefore, UGC is a useful tool for building and promoting destination image [67].

Although research on the role of social media in destination marketing is growing, it evaluates online destination image (ODI) posted on Facebook, Twitter, YouTube, and TripAdvisor [26]. Travel blogs display authentic reality that is not commercially manipulated and staged [68]. Blogs are powerful storytelling sites in comparison to Facebook [69]. Blogs contain the most personal evaluations of a tourist's own experiences, wider range of emotions and richer descriptions of a place [24]. Blogs are ranked the second most important source of destination choice and travel product (and service) consumptions after friends and family [70]. It is a powerful electronic word-of-mouth (eWOM) and a source of organic ODI [71]. Compared to average tourists, bloggers are experienced in travel and are more attached to a destination [72]. Scholars have shed lights on how blogs shape tourist experiences, create meanings and tourist identity [27,73]. This includes the possibility for potential tourists to access tourists' feelings, perceptions, and reflections through stories published in blogs [25].

Bloggers reconstruct and idealise their travel stories by combining memories and emotional experiences, facts and stories about a place using different linguistic techniques,

writing styles, descriptive and emotional words [19]. Emotional experience is how a feeling towards a destination is shaped by an individual's evaluation of the world [16]. Through narratives, bloggers make sense of what they find different from their home culture and construct an imagination known as "rhetorics of othering"—an encounter with the exotic local people [74]. Descriptive (to show differences) and emotional (to imply shock, surprise, or empathy) words are used to express their experiences [19]. As bloggers reconstruct their memories into narratives, they are regenerating the life of their travel experience [73] while extending their emotional experience to others. In other words, bloggers are not only assigning meanings and attachment to a destination, but also projecting their imagination and emotional experiences on others.

Blogger's narratives, whether in the form of written text or visual media, frame the way others make sense of places [75]. Bloggers incorporate lived experiences and knowledge and make sense of a destination by reinterpreting and retelling stories [76]. Through this incorporation, blogger's narratives take on new meanings for a destination and thus narratives are place-making tools [77]. Values and meanings of travel experiences such as stories of risk and challenge, accounts of learning and reflection, accounts of novelty and difference, accounts of self-expansion, and stories of escape are embedded in narratives to allow access to a blogger's emotional experience about a destination [25]. For example, stories of risks taken (such as bungee jumping and skydiving) and challenges (including hardships and difficulties) that dominate blogger's narratives [19] will stimulate reader's imagination and form an organic (unintended exposure to destination) or induced (intended search of destination information) image in the mind of readers. Blogs can generate consequences and risks that challenge the intended destination image [71]. Narratives shared in the post-consumption stage are blogger's perceived images known as complex image. Blogger's complex image shapes potential tourists' pre-visit image, that is, organic and induced images which affect their destination choice [71]. Through destination image, blogs allow tourists to define these meanings of destination and share the meanings with readers. Through narratives, bloggers develop emotional attachment to destinations for their readers.

## 3. Method

Based on the methodological assumptions of this study, mainly an acceptance of multiple realities and the need to immerse oneself in the study, or researcher as instrument, this study assumed an interpretivist inquiry paradigm. Therefore, a qualitative research design enabled this exploratory study to focus on social meanings created by people through text and other forms of communication [78]. The nature of the research question, "how do bloggers' emotional experiences produce SoP?", deemed it exploratory and suitable for a qualitative method that allowed data collection from naturally occurring data [79]. This study employed a QCA to explore tourists' SoP. QCA is a systematic coding process used to subjectively interpret a social phenomenon extracted from a content [80]. To analyse huge amounts of blog data, this study used QCAmap, a scientific software developed by Mayring and Fenzl [81] which deals with texts. QCAmap is guided by strict procedures with qualitative steps that assign passages into categories and quantitative steps that analyse frequencies of categories [82]. QCAmap was used to analyse text data in blogs for a few reasons. First, software such as MAXQDA cannot fully achieve the central content-analytical rules for summary and coding guidelines [82]. Secondly, QCAmap can keep the central content-analytical rules such as category definitions constant throughout the analysis [82]. Next, words or phrases used by bloggers differ across each individual. Words that mention wildlife that belong to the theme "nature" could mean an "adventure" experience. Software such as Leximancer that summarizes the frequency of text linkages may categorise text into a wrong theme because it may not fully capture the meaning. Considering these reasons, QCAmap was chosen as it is effective in capturing meanings in qualitative UGC data.

Specifically, this study used a QCA to explore tourists' SoP to Sabah, Malaysian Borneo, to conceptualise key themes of the region's ODI. Specifically, blogs containing narratives of the region were collected from Google and Bing from December 2021 to February 2022 through purposive keywords search of "Borneo" and "blog". Other purposive sampling criteria include blogs containing post-visit content, written in English as Borneo is known to the English-speaking world and this market is a key segment for Sabah, both unsponsored and sponsored trips. Narratives were thematically coded using QCAmap to reveal key themes for the ODI of Sabah. A coding process was established to enhance trustworthiness as part of the audit trail. Peer review [83] between authors acted as another form of credibility check.

*3.1. Data Collection*

This study relied on a purposive sample of 34 blogs written by Western bloggers that met the sampling criteria. Table 1 depicts these blogs: each blog had minimum one entry or blogpost on this destination, with the highest entries at 14.

**Table 1.** List of blogs.

| No. | Blog Name | Origin | Number of Blogposts |
|---|---|---|---|
| 1 | Charlotte Plans a Trip | The Netherlands | 2 |
| 2 | Not Another Travel Blog | UK | 3 |
| 3 | The Planet D | Canada | 3 |
| 4 | WanderlustingK | US | 1 |
| 5 | Cycloscope | Italy | 4 |
| 6 | The Family Freestylers | UK | 8 |
| 7 | Globe Guide | Canada | 3 |
| 8 | The Globetrotter GP | UK | 2 |
| 9 | Wapiti Travel | The Netherlands | 2 |
| 10 | Meander with Meg | UK | 5 |
| 11 | Ditch the Map | US | 1 |
| 12 | Vegg Travel | UK | 3 |
| 13 | Beard and Curly | US | 2 |
| 14 | The Global Wizards | Belgium | 2 |
| 15 | One Step 4ward | Iceland | 2 |
| 16 | Camels and Chocolate | US | 6 |
| 17 | Bubbles Underwater & Beyond | French | 14 |
| 18 | Flying the Nest | Australia | 2 |
| 19 | No Back Home | US | 1 |
| 20 | Be My Travel Muse | US | 2 |
| 21 | Amalia Explores | US | 3 |
| 22 | No Hurry to Get Home | Mexico | 5 |
| 23 | Amateur Traveler | US | 2 |
| 24 | Wanderlust Movement | South Africa | 2 |
| 25 | Bespoke Bride | UK | 1 |
| 26 | Away We Go | US | 3 |
| 27 | The Happy Kid | Romania | 4 |
| 28 | Mumpack Travel | Australia | 7 |
| 29 | One Million Places | German | 5 |
| 30 | Places of Juma | Austria | 1 |
| 31 | The Martha Blog | US | 5 |
| 32 | Divergent Travelers | US | 4 |
| 33 | Experimental Expats | US & Canada | 3 |
| 34 | The Budget Savvy Travelers | US | 3 |

In total, 34 blogs posted 116 entries on Sabah. All 116 blogposts were saved into a word file for data analysis. The demographic data is insightful. Nine bloggers are European, six from the United Kingdom, sixteen from the U.S.A, two Australian, and one South African. They either travelled solo (11), with a partner (14), with family (seven), or in group (two). Most bloggers paid for their own expenses. Only six were hosted and sponsored by either the press, resorts, travel agencies, or tourist boards. Affiliate links were posted in the content and declaration of unbiased opinions and recommendations were included for sponsored trips (press trips, partnerships). Therefore, the content is considered objective and valid for analysis. Trustworthiness was addressed through engagement with bloggers (through email communication), peer debriefing, purposive sampling, and audit trail of data collection and analysis (coding), and persuasiveness of blog content [84].

### 3.2. Data Analysis

Analysis was completed with QCAmap, a research question oriented analytical software with systematic steps. These steps are illustrated in Figure 3.

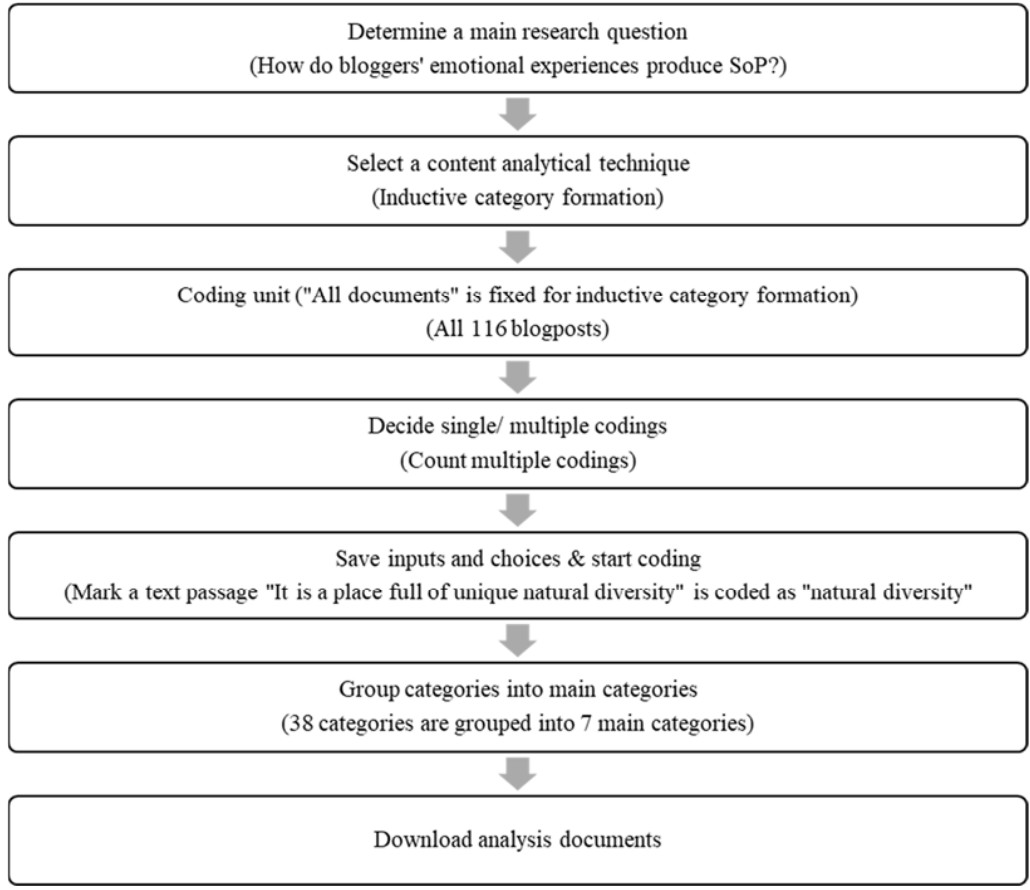

**Figure 3.** Steps of QCAmap.

QCAmap began with determining a main research question. Once a main research question is determined, inductive category formation is selected as the content analytical technique. This technique is selected because it derives "context theories from linguistics" [82] (p. 2). For coding unit, the category "all documents" is fixed for this technique [82], which means that all 116 blogposts are the coding unit. Next, "count multiple codings" is selected because a category that was coded several times within one document (one blogpost) will make sense in a phrase that refers to more than one category. For example, the phrase "*With the history, culture, food and wildlife on offer in Borneo I know I'll be back for more.*" (Blogpost 1 from Meander with Meg) was coded three times as "RQ1-20: Explo-

ration", "RQ1-12: Natural diversity", and "RQ1-41: Cultural diversity". Once these inputs and choices are saved, coding was started by marking a text passage. For example, "*It is a place full of unique natural diversity*" (Blogpost 1 from Amalia Explores) was coded as "RQ1-12: Natural diversity". The coding process generated 38 categories that were further categorised into seven main categories (see Table 2). Once all categories are grouped into the main categories, QCAmap will generate analysis documents.

**Table 2.** Categorisation of sub-categories into main categories.

| Main Category | Category ID | Category Name | Absolute Count | % of SUM |
|---|---|---|---|---|
| Nature | | | 419 | 48 |
| 1 | RQ1-1 | Ancient, oldest rainforest | 16 | 1 |
| 2 | RQ1-12 | Natural diversity | 47 | 5 |
| 3 | RQ1-13 | Home to endemic species | 54 | 6 |
| 4 | RQ1-14 | Pygmy elephant | 8 | 0 |
| 5 | RQ1-15 | Wild, untamed Borneo | 4 | 0 |
| 6 | RQ1-21 | Wildlife | 78 | 8 |
| 7 | RQ1-22 | Unique landscapes (waterfalls, mountains, beaches, islands, diving sites) | 112 | 12 |
| 8 | RQ1-23 | Orangutan | 36 | 4 |
| 9 | RQ1-28 | Proboscis monkey | 12 | 1 |
| 10 | RQ1-29 | Wildlife in natural habitat | 27 | 3 |
| 11 | RQ1-34 | Quiet, peaceful, secluded natural environment | 18 | 2 |
| 12 | RQ1-35 | Turtle | 4 | 0 |
| 13 | RQ1-39 | Sun bears | 3 | 0 |
| Adventure | | | 210 | 24 |
| 1 | RQ1-7 | Hard adventure | 39 | 4 |
| 2 | RQ1-19 | Soft adventure | 94 | 10 |
| 3 | RQ1-20 | Exploration (culture, environment, wildlife) | 77 | 8 |
| Culture | | | 47 | 5 |
| 1 | RQ1-41 | Cultural diversity | 26 | 2 |
| 2 | RQ1-18 | Hospitable locals | 7 | 0 |
| 3 | RQ1-26 | Local way of life | 3 | 0 |
| 4 | RQ1-38 | Cultural differences | 1 | 0 |
| 5 | RQ1-42 | Historical influence | 10 | 1 |
| Environment | | | 72 | 8 |
| 1 | RQ1-3 | Deforestation | 11 | 1 |
| 2 | RQ1-11 | Habitat loss | 11 | 1 |
| 3 | RQ1-27 | Oil palm plantation | 29 | 3 |
| 4 | RQ1-32 | Pollution (rubbish, trash, garbage) | 6 | 0 |
| 5 | RQ1-33 | Threats of poachers, predators, and others | 14 | 1 |
| 6 | RQ1-40 | Tourist's behaviour | 1 | 0 |
| Conservation | | | 34 | 3 |
| 1 | RQ1-4 | Offer solutions for conservation | 11 | 1 |
| 2 | RQ1-16 | Urge for support and conservation | 15 | 1 |
| 3 | RQ1-36 | Helping the wildlife and environment | 8 | 0 |
| Education | | | 11 | 1 |
| 1 | RQ1-6 | The importance of conservation education | 3 | 0 |
| 2 | RQ1-30 | Important information when viewing wildlife | 8 | 0 |

Note: RQ1-8 (Declaration of partnerships and sponsorships) and RQ1-25 (State affiliate programs) are coded for the category "Declaration of unbiased opinions and recommendations", should not be counted as a coding for the categorisation. RQ1-2 and RQ1-17 were deleted for coded repeatedly. RQ1-5 (Local's effort on conservation), RQ1-9 (Food), RQ1-10 (Modernity), RQ1-24 (Information on travel restriction and safety), RQ1-31 (English-speaking locals), and RQ1-37 (Other interesting places) are coded for the category "Unique Themes" that are not subject of this paper, should not be counted as a coding for the categorization.

## 4. Findings

Table 2 presents the main categories and all sub-categories identified in 116 blogposts as well as the absolute word or phrase count for each category. In particular, "nature", "adventure", "culture", "environment", "conservation", "education", and "unique themes"

were identified as the seven main themes representing Western tourists' perceived images of Sabah, Malaysian Borneo. Seven tables with examples of narratives are presented next.

"Nature" is the main theme in the emotional narratives of bloggers who visited Sabah, Malaysian Borneo. It describes the inherent features of Borneo including wildlife, biodiversity, and unique landscapes. Narratives in Table 3 revealed multiple emotional sentiments and phrases, such as "amazement", "peaceful", and "marvellous encounters in the natural environment". Bloggers also used capital letters to express their emotions.

**Table 3.** Emotional Narratives on Theme Nature.

| Blog | Narratives |
|------|-----------|
| One Step 4ward | Blogpost 1: *it was possible to see Orangutans in the Borneo IN THE WILD!* |
| Divergent Travelers | Blogpost 1: *There is just something about the sound of nothing, the sound of distant birds and crashing monkeys that just makes me feel at peace. Borneo delivered on this aspect.* |
| Bespoke Bride | *The sound of silence was suddenly broken by the sound of a pygmy elephant trumpeting in the distance. I will never forget that sound, it took me a little while to realise that it was in fact a real live elephant in the wild and not something I was watching on the TV from the comfort of my own home.* |

"Adventure" emerged as the second important theme of emotional narratives among the bloggers. This theme includes unusual and existing activities found in the natural environment, such as hard adventure, soft adventure, and exploration. Emotional sentiments including "difficulties", "hardest", "thrilling", "surprise", "excited", and "lucky" were used to describe bloggers' emotional encounters (see Table 4).

**Table 4.** Emotional Narratives on Theme Adventure.

| Blog | Narratives |
|------|-----------|
| Bubbles Underwater & Beyond | Blogpost 1: *my pulse accelerates, when I plunge into the shadows, then into the intimidating darkness of this rather vast cave whose bottom we do not see.* |
| Globe Guide | Blogpost 3: *This is by far the most difficult part of the climb, and sadly people have been killed on this leg after tumbling off the extremely steep mountainside.* |
| Places of Juma | *we did thrilling guided jungle walks and guided river cruises.* |
| Camels and Chocolate | Blogpost 3: *found a mama leatherback in the middle of laying eggs!* |
| Charlotte Plans a Trip | Blogpost 2: *On our way back to the field centre we see different kinds of hornbills and a group of monkeys crossing the other side of the road by jumping from one tree to another tree. It almost looks like the monkeys can fly too. We realize how lucky we are to be seeing all those animals.* |

"Environment" is the third most emotionally narrated theme of bloggers. This theme is related to threats to natural environment including development activities that affect the wildlife habitats and environmental pollution. From Table 5, it is seen that bloggers use rather strong phrases to describe their disappointment, surprise, and sadness towards the environment of the region. Notably, the narratives are not supported by empirical evidence historical accounts.

**Table 5.** Emotional Narratives on Theme Environment.

| Blog | Narratives |
|---|---|
| Divergent Travelers | Blogpost 1: *I don't know what it is with Malaysia, but they don't give a shit about picking up their garbage. We saw garbage everywhere in Penang and again in Sabah.* |
| Not Another Travel Blog | Blogpost 1: *We were truly shocked to see mile after mile of palm oil plantations on what would have been rainforest as we travelled through Borneo.* |
| Meander with Meg | Blogpost 3: *Sadly, deforestation of their habitat to make way for palm oil plantations, timber and settlements has seen the proboscis monkey population decline rapidly and they are now an endangered species. They were also hunted for food and for traditional Chinese medicine.* |

The next most important theme is "culture", which is the manifestations of a collective group of people who inherently live in Borneo. This theme includes local arts, tribes, traditions and religion. From 47 narratives, emotional sentiments such as "fascinating" and "amazement" were identified. Table 6 lists the emotional narratives on theme culture.

**Table 6.** Emotional Narratives on Theme Culture.

| Blog | Narratives |
|---|---|
| Mumpack Travel | Blogpost 5: *it's such an amusing way to learn about Borneo's fascinating tribal history, customs, food and religion in a fun and interactive tour of villages.* |
| No Hurry to Get Home | Blogpost 1: *The Sabah region, in general, is a melting pot of cultures and backgrounds, and its gastronomy is no different.* |
| Amateur Traveler | Blogpost 1: *Sabah is a fantastic place. . . . depth of culture with 37 indigenous races of people. And one of the reasons I live here, it's very safe. The people that are the big draw. They're wonderful people, very hospitable. They're very naturally hospitable, very friendly, very willing to help.* |

"Conservation" is the fifth theme found in bloggers' narratives. This theme is related to the action of protecting, preserving, conserving, and encouraging others to conserve the natural environment and wildlife of Borneo. Bloggers would offer solutions for conservation through arousing reader's conscious or urge for support and conservation (see Table 7):

**Table 7.** Emotional Narratives on Theme Conservation.

| Blog | Narratives |
|---|---|
| Amalia Explores | Blogpost 3: *We can all make more conscious decisions about our consumption of palm oil.* |
| The Budget Savvy Travelers | Blogpost 1: *we were reminded that the best way to preserve the precious rainforest and support wildlife conservation is through eco-tourism.* |
| The Family Freestylers | Blogpost 8: *Incredibly, this is one of only two places on earth where 10 primates species can be found together, including the orangutan, the proboscis monkey and the Bornean gibbon. Home to 250 bird species, 50 mammals, 20 reptiles species and 1056 plant species, the biodiversity here is mind-blowing. It's also argued to be the last forested alluvial floodplain in Asian. Lets just say it's a very special place on earth and it needs to be protected.* |

The sixth theme is categorised as "education", which is associated with the process of receiving or giving systematic instruction about conservation and preservation of the natural environment and endangered wildlife. Although there were only 11 phrases on

this theme, bloggers are seen trying to educate readers on the importance of conservation through reminding others to be a responsible tourist (see Table 8).

**Table 8.** Emotional Narratives on Theme Education.

| Blog | Narratives |
| --- | --- |
| Divergent Travelers | Blogpost 2: *This is not a zoo and sightings of the Orangutans is not guaranteed.* |
| The Global Wizards | Blogpost 1: *The reason you come here is to* **spot wildlife** *of course. But adjust your expectations. Don't expect you'll be able to check everything on your bucket list. These animals are wild, which means they don't pop up every time you want them to. For example, we didn't see the Pygmy Elephants. However, no need to worry! You'll see plenty of wildlife!* |

*Capturing Attributes That Are Meaningful to Bloggers*

The findings revealed six themes representing Western tourists' perceived images of Sabah, Malaysian Borneo. First, "nature" is the main attribute that is emotionally valued by all bloggers. This supports [28–34] who found Western tourists arrived in Borneo with the expectation of seeing the wild and untamed Borneo. Bloggers are emotionally attached to the "nature" of Sabah through emotional sentiments such as amazement, pleasure, peaceful, happiness, joy, and appreciation. Orangutan is the symbolic icon representing "wildlife" in Borneo [30] with 36 counts. Other wildlife frequently mentioned are proboscis monkey (12 counts), pygmy elephants (8 counts), turtle (4 counts), and sun bears (3 counts). Secondly, "adventure" is found to emerge from human-environment interaction. As bloggers interact (emotionally) with the natural environment, emotional sentiments such as thrilling, exciting, scary, lucky, and surprised describe their engagement with exploration and adventure activities. Thus, "adventure" is a result of human-environment interactions.

While Borneo is renowned for its nature, adventure, and culture [28–34], the third most meaningful attribute of Sabah among bloggers in this study is not "culture". Bloggers are found to be emotionally involved and concerned about the "environment" of Borneo. Emotional sentiments and words like sadness, disappointment, doubt, and shocked were used to describe bloggers' emotions about the environment. "Culture" appears as the fourth most important attribute among the bloggers, which supports past studies [28–34]. Emotional sentiments such as fascinating and amazement describe how bloggers' value their cultural experiences.

Besides the four main themes, the findings also revealed two attributes which are meaningful to bloggers. These two themes labelled as "conservation" and "education" emerged from bloggers' emotional engagement to Borneo, which is their concern towards the "environment". Narratives depict concern for environmental issues such as deforestation and pollution, which threaten the natural environment's sustainability and wildlife habitat; emotional sentiments that encourage and support the "conservation" including protect, preserve, support were used in 20 blogposts. For "education", teaching, educating, instructing, and other encouraging phrases were emotional sentiments used most frequently in written narratives.

Inherently, the experience of travel provides emotional engagement with a destination. However, we propose that for bloggers, the act of first experiencing Sabah, and secondly documenting it in narrative form as a blogpost, evokes a deeper experience. The QCA identified six attributes across the 116 blogposts that shape readers' experience and perceptions [4] about this destination. Bloggers SoP captured in the narratives now guide others to sense of places before visitation [36].

**5. Discussion**

The purpose of this study was to explore how bloggers emotional experiences produce SoP for potential tourists. For this, a QCA was conducted on a purposive sample of blogposts for Sabah. The emotional sentiments identified through narratives were catego-

rized based on place-related attributes. Place-related attributes may reflect the relationship between tourists and places as well as the type of attachment to a location [55–57]. This discussion seeks to demonstrate that the emotional experiences shared on blogs do produce a secondary SoP that would entice potential tourists to visit.

Jarratt et al. [58] suggest one way to identify SoP is through place-related attributes and promotional images. When developing a SoP toolkit for Morecambe Bay, Jarratt et al. [58] employed a case study approach and used images and postcards to evoke personal experience and memories of tourists. In their views, the relationship between tourists and places involves an emotional relationship and produces a SoP that contains the essential emotional aspects of a successful destination brand [58]. In this study, bloggers' modified-induced or complex image formed in the post-visitation stage [37,38] demonstrates an emotional bond between bloggers and Sabah and produces SoP through the narratives that focus on key attributes, evoking meaningful emotions for the reader or potential tourist.

*Subjective Interpretation That Guides Potential Tourists' Pre-Travel Experience*

Bloggers make sense of places as they retold stories and emotional experiences in their narratives [19]. In this context, bloggers assigned meanings to places, hence SoP is rooted in narratives. Bloggers reproduce stories of Sabah based on their personal and subjective interpretation and produced a modified-induced or complex image for themselves. These post-visitation images derived from the emotional interaction between bloggers and the destination represent place meanings, which are bloggers' SoP towards Sabah. Bloggers' narratives then allow potential tourists to produce subjective interpretations of the location. The narratives provide information about the destinations, allow potential tourists to generate subjective interpretations, and evoke an emotional engagement [18]. When potential tourists read blogs, their imagination of the place will form an organic or induced image of the destination and emotional engagement with the place. These images formed through the narratives of bloggers [39] will help potential tourists make sense of Sabah and form a SoP within potential tourists in the pre-travel stage [36]. The next section illustrates how bloggers' narratives demonstrate an emotional relationship between bloggers and place and potentially form organic or induced images in the mind of potential tourists.

The findings show that the six attributes are interconnected. "Nature" is the main attribute that made bloggers emotionally attached to Sabah, Malaysian Borneo. It creates "adventure" opportunities to explore, consists of diverse fascinating cultural groups ("culture") [35], and generates "environment" concerns such as deforestation and pollution threaten wildlife. As "nature" provides certain values that bloggers appreciate and grow attached to, in return, bloggers aim to "conserve" and "educate" potential tourists so that the main value of Sabah, "nature", can be returned to the original state—the wild and untamed Borneo filled with ancient rainforest, unique landscapes, and endemic wildlife. Based on the analysis, this study proposes a tourists' SoP model based on bloggers' narratives (perceived image), which is illustrated in Figure 4.

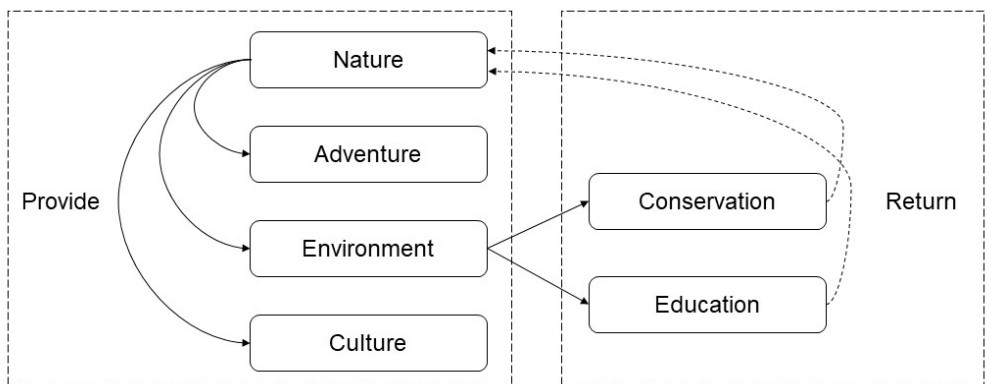

**Figure 4.** Proposed Tourists' SoP Framework.

The six key destination images were formed through bloggers' knowledge and emotional interactions with the destination. Before the trip, bloggers developed emotions and connection with Sabah based on its historical link with the European literature [28]. Nature-based tourist destination is the meaning assigned or bloggers' SoP for Sabah pretravel. During the trip, bloggers view what they see as threats to natural environment and are concerned that these threats will alter their pre-visit images acquired from the literature and tourist brochures. As reality contrasts bloggers' expectation, their emotions towards the destination deepened as they perceived the loss of SoP, which, in this case, is Sabah is not at all wild Borneo but a nature-based destination with opportunities for adventure and cultural exploration that needs environmental conservation. In this stage, bloggers developed awareness to protect the destination's main image, which is a way to protect their original SoP. In the post-travel stage, bloggers reconstructed their travel stories using pre-travel knowledge, experiences during travel, and post-travel reflection. This includes the use of emotional words and different linguistic techniques [19] to make sense of what they found different from their expectations. In this context, emotional words from amazement (with the nature, adventure, and culture) to disappointment (with the threats), conservation (of environment), and education (on conservation) were values and meanings assigned by bloggers to Sabah. Bloggers value the region as a nature-based destination that needs conservation. This shows that tourists are emotionally attached to a destination [17]. These six perceived images, narrated through emotional tones, will potentially produce an organic image for those who are unaware of the destination, and an induced image for those who search for travel information. For those whose SoP is influenced by early literature and tourist brochures, bloggers narratives will induce an emotional engagement in potential tourists, especially those who prefer nature-based tourist destinations.

This study used text data from blogs to develop a SoP model for Sabah. Bloggers provide subjective evaluations with a wide range of emotions [24] such as amazement, anticipation, joy, excitement, belonging, frustration, dread, sadness, and appreciation. These emotions show that bloggers are more attached to a destination compared to other tourists [72]. Although some emotions express anger and frustration, they are used to encourage conservation efforts [40]. Three conclusions can be made in this study. First, bloggers' SoP can be revealed through blogs. Similar to the findings of previous research [43,58,59], this study found that tourists' SoP can be captured through emotional sentiments reflected in blogs. In addition, blogs have demonstrated meaningful social values that are useful for the place branding of Sabah, which supports the views of McCreary et al. [10]. Secondly, tourists' SoP evolves with time and spaces, which is in line with the views of Wang et al. [65]. In the past, Borneo was renowned for three main themes, namely "nature", "adventure", and "culture". As time passes, tourists' preferences evolved from pure nature appreciation to environmental concern. As their concerns heighten, "environment", "conservation" and "education" emerged as new themes for Western tourists' SoP. Finally, SoP and place brand are dynamic concepts that can be developed across experiences of individuals who are emotionally attached to the place. As such, this study agrees with Jarratt et al. [58] on the fluid nature of places that causes the dynamism of SoP and place brand. To construct a SoP model for any destinations, subjective and emotional interaction between people and place should be considered [2]. A successful destination brand emphasises the degree of congruity between brand attributes and tourists' perceived images [58]. Tourism providers and destination marketers should pay special attention to the themes of "environment" and "conservation" to further develop sustainable efforts in preserving nature and regenerative tourism [85] efforts in Sabah, Malaysian Borneo, to attract tourists who are more concerned about sustainability in their travel destinations. Tourism value could be created by emphasising environmental and natural preservation in tourism sites, which would contribute favourably to tourism well-being and as a give back to the tourism destinations.

Finally, this study responded to the call of Ruiz-Real et al. [13] and Nelson et al. [42] and developed a SoP model for Sabah, Malaysian Borneo, that shows how bloggers are emotionally attached to a place. The SoP model shows that bloggers are emotionally

attached to the main attribute of a place. Therefore, the SoP model developed is this study adds to the literature of destination branding from the perspectives of tourists.

## 6. Conclusions and Future Research

Through a qualitative approach, this research has developed a SoP model for destination attachment. Through analysing blogger's emotional experiences narrated in blogs, six key perceived images were identified as a SoP among the tourists. Four perceived images, namely nature, adventure, environment, and culture are in line with the findings of previous studies [28,29,31–34,40], and two additional themes, namely conservation and education, were identified. The findings show that these six themes are interconnected. First, the theme "nature" indeed offers opportunities to explore "adventure" and "culture", which support the findings of Markwell [35]. Secondly, theme "environment", which derived from theme "nature", also offers opportunities for environment "conservation" and conservational "education". As "nature" provides other key images that meet tourists' pre-travel SoP, tourists developed a new SoP, namely "environment", based on their concern towards their original SoP. This newly constructed SoP then leads tourists to perceive the need to give back by creating two additional SoP, which are concerned with protecting the natural environment and educating others to behave responsibly.

This study builds on previous studies that suggest tourists' SoP can be assessed through UGC data [10], particularly in exploring the emotional bonds between bloggers and Sabah [23]. Travel blogs contain tourists' emotions [24], which allow readers to develop a pre-travel SoP. Accessing tourists' SoP will allow a destination brand to create a strong destination image that evokes emotional engagement between potential tourists and the destination [18]. This study indicates how tourists are emotionally attached to a destination post-travel through narratives [42]. A tourist destination bond is developed based on how tourists value a place. Bloggers' sense of Sabah, Malaysian Borneo, although predetermined through literature and tourist brochures, is deepened and redeveloped as they recalled and retold stories in their blogs [60]. The findings of this study add to and clarify the knowledge of the creation of destination attachment and SoP in tourism literature by emphasising the value of emotional narratives shared by bloggers.

This study is not without limitations. First, this study developed a tourists' SoP model using a rather unique case. As Jarratt et al. [58] argued, a tourists' SoP model can only be developed according to local features. The model developed in this study may only be appliable to other destinations with similar features. Secondly, blogs written in other languages were not considered in this study. Sabah welcomes tourists from all around the world, including the Philippines, Indonesia, Japan, South Korea, German, and France. Blogs written in other languages could provide different perspectives on SoP. Next, this study only used text data for the analysis. Videos and photos shared in the blogs can provide additional emotional data useful for the development of tourists' SoP [18]. Future studies should identify SoP based on the experiences of Asian tourists. In future investigations, the themes highlighted in this study could be examined for their relative influence on tourist inclinations to return and promote the place to others.

**Author Contributions:** S.F.W.: Conceptualization, Methodology, and Formal Analysis, Data curation, Writing—Original draft preparation; B.K.K.: Conceptualization, Methodology, Writing—Reviewing and Editing, Supervision, Validation, Visualization; B.K.B.: Writing—Reviewing and Editing, Supervision, Validation, Visualization. All authors have read and agreed to the published version of the manuscript.

**Funding:** This research received no external funding.

**Institutional Review Board Statement:** Not applicable.

**Informed Consent Statement:** Not applicable.

**Data Availability Statement:** Not applicable.

**Conflicts of Interest:** The authors declare no conflict of interest.

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
