# Peer review of "Sense of Place: Narrating Emotional Experiences of Malaysian Borneo through Western Travel Blogs"

_tourismhosp, doi:10.3390/tourhosp3030041_

Round 1

Reviewer 1 Report

Dear authors,

This is an interesting and well-written paper. Overall the manuscript in its current form does a good job of engaging with sense of place literature in the context of Malaysian Borneo as they are presented through western travel blogs.

There are, however, a couple of issues I urge you to address. There is a lack of clarity on your "interpretivist position" and how this has informed your analysis. The discussion of the method should include a discussion of how the chosen methods fit within your ontological and methodological position. The interpretivist paradigm should also be illustrated in the discussion and analysis of your findings. This would include further interrogation of the categories in your model, and an exploration of their meaning (eg. culture, adventure etc.).  This discussion should help explore the complex, open, dynamic and contested nature of places (See Lichrou and Panayiotopoulos, 2021) and the emotional dimension of sense of place (Jepson, D., & Sharpley, R. (2015). More than sense of place? Exploring the emotional dimension of rural tourism experiences. Journal of Sustainable Tourism23(8-9), 1157-1178.)

All my best and good luck with your manuscript.

Author Response

Dear Reviewer 1,

Thank you for the valuable comments and suggestions. We responded to your comments by making major revision to the manuscript, please see attachment (revised manuscript). 

Thank you.

Reviewer 2 Report

thank you for submitting your paper to JTH. this is an interesting paper and an equally interesting paper. I feel that the paper addresses a relevant topic for academia, one that still needs firther exploration. The analysis via the blogs, the idea of looking at tourist attachment are all pertinent. I feel that the paper could be improved by following those recommendations:

I find that the introduction could be clearer. the argumentation for the shift from residents to tourists in regard to SOP needs to be more clearly set out. For example, the second paragraph starts with two sentences referring to residents, then tourists and this is unclear. The first paragraph on mage 2 is also confusing. please be very clear about what you aim to achieve, which variables you are using in your study and how are those variables related/interconnected. 

line 70: blogs who have a wider range of emotions attract less attention? Please explain.  And if it is the case, why choose blogs for this study? 

Page 4-112: please define "induced", I guess it refers to Gunn's model, please state then. 

Throughout your literature review, please make sure you clearly differentiate all the concepts addressed. 

Methods: please rewrite the justification for QCAmap more precisely, not every reader will be familiar with it. 

The discussion is the most problematic part of the paper, I am afraid but I failed to identify how the results identify SOP? In the first line the authors mention "perceived image" and indeed, this is what those results do.  The results list a range of features that are characteristic of the destination, nature, environment.. it identifies the key dimensions of the image or experience if you prefer, but I failed to see how those results were associated to the notion of place attachment. There is indeed the place of emotions, but still I did not identify how those emotions were connected to sense of place. I think the demonstration need to be strengthened for the whole paper to make sense. It is a good study, the authors understand the concepts they are dealing with, but the demonstration is lacking, please revise your findings so that this demonstration becomes more apparent and convincing in the discussion. 

There are some english/grammar mistakes please have the text read over by a native speaker. Also, if you decide to use the term SOP, please do so throughout the paper. 

Author Response

Dear Reviewer 2,

Thank you for the valuable comments and suggestions. We responded to your comments by making major revision to the manuscript, please see attachment (revised manuscript), as well as the point by point responses below. 

Thank you.

Reviewer 2: Thank you for submitting your paper to JTH. this is an interesting paper and an equally interesting paper. I feel that the paper addresses a relevant topic for academia, one that still needs further exploration. The analysis via the blogs, the idea of looking at tourist attachment are all pertinent. I feel that the paper could be improved by following those recommendations:

Point 1: I find that the introduction could be clearer. the argumentation for the shift from residents to tourists in regard to SOP needs to be more clearly set out. For example, the second paragraph starts with two sentences referring to residents, then tourists and this is unclear. The first paragraph on mage 2 is also confusing. please be very clear about what you aim to achieve, which variables you are using in your study and how are those variables related/interconnected.

Response 1: Thank you for pointing out the confusion. A brief argumentation for residents’ SoP were added in “Introduction” (page 1, paragraph 2) and the sentences referring to tourists were moved to the next paragraph. Revision is shown from page 1-3.

Point 2: line 70: blogs who have a wider range of emotions attract less attention? Please explain.  And if it is the case, why choose blogs for this study?

Response 2: For line 70, according to Mariani (2020), blogs contain wide range of emotions that are accessible to potential tourists receive less attention in academia compared to other types of social media (Facebook, Twitter, YouTube, and TripAdvisor). Explanation for why blogs are chosen for this study is added afterwards. Please see page 2, paragraph 3.

Point 3: Page 4-112: please define "induced", I guess it refers to Gunn's model, please state then. Throughout your literature review, please make sure you clearly differentiate all the concepts addressed.

Response 3: Thank you for the reminder. The term is defined and explained in page 4.

Point 4: Methods: please rewrite the justification for QCAmap more precisely, not every reader will be familiar with it.

Response 4: Added the definition for QCAmap in page 8, Section “3. Method”.

Point 5: The discussion is the most problematic part of the paper, I am afraid but I failed to identify how the results identify SOP? In the first line the authors mention "perceived image" and indeed, this is what those results do.  The results list a range of features that are characteristic of the destination, nature, environment.. it identifies the key dimensions of the image or experience if you prefer, but I failed to see how those results were associated to the notion of place attachment. There is indeed the place of emotions, but still I did not identify how those emotions were connected to sense of place. I think the demonstration need to be strengthened for the whole paper to make sense. It is a good study, the authors understand the concepts they are dealing with, but the demonstration is lacking, please revise your findings so that this demonstration becomes more apparent and convincing in the discussion.

Response 5: Thank you for your constructive comment. A clearer demonstration of how bloggers’ emotions were connected to sense of place was revised in the findings and discussion from page 13-21.

Point 6: There are some english/grammar mistakes please have the text read over by a native speaker. Also, if you decide to use the term SOP, please do so throughout the paper.

Response 6: Revised the English/grammar mistakes and the term throughout the paper, thank you!

Round 2

Reviewer 2 Report

The authors have made some effort to improve their paper. I am still no totally convinced with the argument but the link between emotions and SOP has been strengthened thank you.